# Modification of Poly(lactic acid) by the Plasticization for Application in the Packaging Industry

**DOI:** 10.3390/polym13213651

**Published:** 2021-10-23

**Authors:** Karolina Gzyra-Jagieła, Konrad Sulak, Zbigniew Draczyński, Stepan Podzimek, Stanisław Gałecki, Sylwia Jagodzińska, Dominik Borkowski

**Affiliations:** 1Lukasiewicz Research Network-Institute of Biopolymers and Chemical Fibres, 19/27 M. Skłodowskiej-Curie Street, 90-570 Łódź, Poland; k.sulak@ibwch.lodz.pl (K.S.); s.galecki@ibwch.lodz.pl (S.G.); s.jagodzinska@ibwch.lodz.pl (S.J.); d.borkowski@ibwch.lodz.pl (D.B.); 2Faculty of Material Technologies and Textile Design, Lodz University of Technology, 116 Żeromskiego Street, 90-924 Łódź, Poland; zbigniew.draczynski@p.lodz.pl; 3Wyatt Technology Europe, 56307 Dernbach, Germany; Stepan.Podzimek@synpo.cz; 4Faculty of Chemical Technology, Institute of Chemistry and Technology of Macromolecular Materials, University of Pardubice, 53210 Pardubice, Czech Republic; 5SYNPO, 53207 Pardubice, Czech Republic

**Keywords:** polylactic acid, plasticization, SEC-MALS, biodegradable polymers, FTIR-ATR

## Abstract

Plastic products, especially in the packaging industry, have become the main commodities penetrating virtually every aspect of our lives. Unfortunately, their omnipresence is not neutral to the natural environment. Pollution in the form of microplastics is a global problem. Therefore, green technologies that enter into the circular economy become an important topic. As part of the research work, the modification of poly(lactic acid) has been studied for use in the packaging industry. Due to its intrinsic rigidity, plasticizing substances had to be introduced in PLA in order to improve its plastic deformability. Both high-molecular compounds such as ethoxylated lauryl alcohol, block copolymer of ethylene oxide and propylene oxide, and ethoxylated stearic acid as well as low-molecular compounds such as di-2-ethylhexyl adipate, di-2-ethylhexyl sebacate, and triethyl citrate were used. The samples extruded from plasticized polymers were characterized using differential scanning calorimetry, thermal gravimetric analysis, and mechanical properties including Young’s modulus. The melt flow rate (MFR) and molar mass distribution were determined. For all modified samples the glass transition temperature, depending on the plasticizer used, was shifted towards lower values compared to the base polymer. The best result was obtained for di-2-ethylhexyl adipate (ADO) and di-2-ethylhexyl sebacate (SDO). The elongation at break increased significantly for ADO at about 21%. The highest elongation was obtained for SDO (about 35%), although it obtained a higher glass temperature. The degradation of the polymer was not observed for both plasticizers. For these plasticizers (ADO and SDO) it also lowered Young’s module by about 26%, and at the infrared spectrum deformation of peaks were observed, which may indicate the interaction of the ester carbonyl group of PLA with plasticizers. Therefore it can be concluded that they are good modifiers. The selected plasticizers that are used in the production of food contact materials, in particular in the production of PVC (polyvinyl chloride) food films, also exhibited great potential to be applied to PLA food films, and exhibit better properties than the citrate, which are indicated in many publications as PLA plasticizers.

## 1. Introduction

Global warming and climate change have been important issues in recent years, raised by the European Union and other world economies. The special report “Climate Change and Land” prepared by the Intergovernmental Panel on Climate Change indicates the current problems that need to be faced and alerts about the dangers to the natural environment. Recently, post-consumer plastic waste in the form of microplastic has become a global problem [1]. It can be found in seawater, freshwater, sewage, food, air, and even in drinking water [2,3,4,5]. In response to the current situation, the European Union (EU) is taking steps to adapt its economy to a “green” future and also to strengthen the EU’s competitiveness. The environmental protection and new consumer laws are the priorities of this plan so that sustainable products will become the standard in all Member States [6]. The EU wants to reduce plastic waste because about 26 million tonnes of plastic waste are generated in the EU every year. Unfortunately, about 30% of it is recycled, some of the waste is exported to non-EU countries for treatment. A large percentage of waste goes to a landfill but the most important problem is that much waste ends up in nature (rivers, seas). Among such activities, there is putting into the market packaging products with a low greenhouse gas (GHG) emission index that fit in the rules of circular economy [7,8]. Therefore, intensive research is being currently carried out with the use of so-called double green polymers that are derived from biomass and undergo biological decomposition processes [9,10]. The introduction of biodegradable materials into everyday use may contribute to the reduction of huge amounts of solid waste, the recycling of which is either unprofitable or impossible [11,12,13,14]. A great part of bioplastic production is consumed by the packaging sector, which makes it worth seeking new, pro-ecological technologies for this industry [15].

Poly(lactic acid) (PLA), belonging to the group of biodegradable polymers obtained from biomass, is currently of great interest in the packaging industry [16,17,18,19]. PLA has good mechanical properties compared to traditional polymers [20,21,22,23,24]. Because the EU is focused on green technology solutions, biopolymers such as PLA have become noticeable by producers from various industries. Despite the higher price of PLA than traditional polymers, the social and environmental aspect becomes more important than the economic cost. In developed or developing countries, food surpluses can be redirected to the plastics sector. PLA could be used in many not yet exploited industrial sectors. It could reduce the generation of large amounts of waste, which often residual and pollute the environment [25,26]. The important concept of extended producer responsibility (EPR) was introduced in the Directive 2008/98/EC on waste and it induces us to look for new solutions using biodegradable polymers. EPR systems in Belgium (Fost Plus), Italy (CONAI), Netherlands (Afvalfonds Verpakkingen), Spain (Eco Embes), Sweden (FTI), and France (CITEO) are the frontrunners, providing financial incentives to companies to move to more recyclable packaging. Other regions of the world also support the EPR system. Chile and Columbia already have EPR laws. South Africa is on the way to making EPR mandatory and small countries e.g., Jordan, Vietnam are also interested in that solution [27]. Therefore, PLA may become an alternative that fits in with the circular economy and the system EPR [28]. Currently, paper is the most commonly used in eco-packaging. Plates, drinking straws, and cups are produced, but the paper used is coated, which makes recycling difficult. During each recycling cycle of the paper, the fibers are shortened and weakened. Therefore, recycled paper has a limited lifetime. Furthermore, it may contain various harmful substances from printing inks or coatings which may disqualify recycled paper for use as food packaging. However, the use of only original raw material is not ecological. Therefore, new solutions with the use of biopolymers such as PLA are being sought.

Extensive studies based, among others, on the product life cycle assessment (LCA) confirm the advantage of PLA over polymers of petrochemical origin [29,30,31,32]. PLA is considered one of the materials that are alternative to polymers obtained from crude oil, which will allow to reduce the consumption of this fossil raw material and lessen the impact on the environment [33,34]. Great interest in this polymer results from its degradation potential in the natural environment [35]. As a result of the biodegradable process, low-molecular products (monomers, dimers) are formed [36,37]. Scientists have proved that higher temperature promotes the degradation of PLA, and the compost microorganisms are more active than soil microorganisms [38,39,40,41].

The possibilities of searching for new applications for poly(lactic acid) are wide but often require some polymer modification which is brittle and stiff. PLA plasticization process can improve its plastic deformability. This is especially important in the packaging industry where products must meet strict functional requirements in order to protect food products. In the packaging sector, new technological solutions are important; therefore, there has been an increase in interest in bio-packaging. Consumers having an interest in organic food and the ecological aspects of food are also interested in ecological packaging. This is a great potential for bio-plastic packaging materials such as PLA. Ceresin experts predict the demand for bioplastic packaging in Europe in the coming years will increase by over 15% annually, which creates great potential for research on the modification of biopolymers including PLA.

The aim of this study was to modify PLA by a plasticizing process in order to improve its plastic deformability. Both high-molecular compounds with ether bonds such as ethoxylated lauryl alcohol, block copolymer of ethylene oxide and propylene oxide, and ethoxylated stearic acid, and low-molecular compounds with ester bonds such as di-2-ethylhexyl adipate (Ergoplast ADO), di-2-ethylhexyl sebacate (Ergoplast SDO), and triethyl citrate (TEC) were used as plasticizers. Ergoplast SDO and ADO are the primary plasticizers giving the PVC (polyvinyl chloride) products very good resistance to low temperatures. In mixtures, PVC has a high plasticizing capacity, low volatility, and good dielectric properties. Therefore, the research was aimed at checking whether the use of these plasticizers has a similar effect on PLA as for PCV. While in the literature, citrates are indicated as good plasticizers for PLA [42,43,44]. The use of low-molecular compounds fits into the free-volume theory. The theory assumes the occurrence of a free internal volume in a polymer for the movement of the polymer chain, which imparts flexibility. Free volume comes from the motions of chain ends, side chains, and the main chain. One way to increase the motions can be the introduction of plasticizer molecules about lower molecular mass into the polymer matrix. It implies the addition of molecules with Tg lower than the Tg of polymers and the relatively small plasticizer molecules, which add a great free volume to the system. The low-molecular plasticizer acts on the amorphous part to increase the flexibility of the polymer and its structure is very important [45,46,47].

## 2. Materials and Methods

The plasticization process was carried out by mechanical mixing in a melted state in a twin-screw extruder. The obtained regranulates were subjected to thermal characterization using differential scanning calorimetry (DSC), thermal gravimetric analysis (TGA), and melt flow rate (MFR). Molar mass distribution was determined using the gel permeation chromatography/size exclusion chromatography (GPC/SEC) technique with multi-angle light scattering (MALS) detection. The obtained regranulates were assessed using a scanning electron microscope (SEM) to observe the changes on the surface. A filament extruded from the obtained regranulates was subjected to mechanical tests including determination of Young’s modulus. The tests made it possible to assess how the chemical structure of plasticizers can influence the intermolecular interactions in the polymer and thus its functional properties. Understanding this relationship will allow for the new applications of PLA, which until now were not possible due to the nature of the polymer.

The polymer used for the research was poly(D, L-lactide) 6201D from Nature Works^®^ LLC (Minnetonka, MN, USA). The polymer characteristics according to the manufacturer’s data are presented in Table 1. This grade of biopolymer can be used in food packaging materials and, as such, is a permitted component of such materials pursuant to Section 201(s) of the Federal, Drug, and Cosmetic Act, and Parts 182, 184, and 186 of the Food Additive Regulations.

The PLA was subjected to the plasticization process using the following substances:1.ethoxylated lauryl alcohol, ROKAnol L80/50 W (PCC Rokita; Brzeg Dolny, Poland); 3700 g/mol2.copolymer of ethylene oxide and propylene oxide, ROKAmer 2950 (PCC Rokita; Brzeg Dolny, Poland); 2950 g/mol3.ethoxylated stearic acid, ROKAcet S 24 (PCC Rokita; Brzeg Dolny, Poland); 1340 g/mol4.bis(2-ethylhexyl) adipate, Ergoplast ADO (Boryszew; Sochaczew, Poland); 370 g/mol5.bis(2-ethylhexyl) sebacate, Ergoplast SDO (Boryszew; Sochaczew, Poland); 426 g/mol6.triethyl citrate, TEC (Sigma-Aldrich; Saint Louis, MO, USA); 276 g/mol

The plasticizers used to modify samples 1, 2, and 3 were macromolecular compounds (nonionic surfactant) containing ether bonds. The research aimed at checking whether non-ionic surfactants reduce intermolecular interactions according to the lubricity theory.

The plasticizer molecules act as lubricants for the polymer chains and reduce their internal resistance to sliding [45,46]. On the other hand, the plasticizers used to modify samples 4, 5, and 6 were low-molecular compounds containing ester bonds. The plasticizing process was carried out by mechanical mixing of PLA containing 10% wt. of the plasticizer in the melted state in a twin-screw extruder (L/D: 420/16) (Zamak Mercator UNIDRIVE 11, Skawina, Poland) equipped with eleven heating zones operating in the temperature range of 200–220 °C (Figure 1). Extrusion was carried out at a screw speed of 150 rpm and torque in the range of 1–2 Nm. A single-orifice forming head was used after extrusion, the filament was quenched in a water bath and then cut into pellets, which were first air-dried at an ambient temperature of approx. 23 °C and then for 48 h in the vacuum dryer (Binder, Tuttlingen, Germany) at 35 °C under reduced pressure (20 hPa).

*Thermal analysis* of samples was carried out by means of DSC (Differential Scanning Calorimetry) using Diamond apparatus (Perkin Elmer, Waltham, MA, USA). The first and second heating scans and the first cooling scan for the polymer were performed in a temperature range of −60–200 °C. The samples were scanned at a heating rate of 20 °C/min.

*The crystallinity degree* of samples was determined using DSC analysis using the following equation: CD (%) = (∆H/∆H_100%PLA_) × 100%
where:
∆H-difference between the enthalpy of melting and cold crystallization the tested sample∆H_100%PLA_-enthalpy of melting for completely crystalline PLA (93.1 J/g) [48,49].

*Thermal gravimetric analysis (TGA*) of samples was carried out using the HI-RES TGA 2950 Thermogravimetric Analyser (TA Instruments, New Castle, De, USA) in a nitrogen atmosphere with a heating rate of 5 °C/min in the temperature range of 20–400 °C.

*Melt flow rate (MFR)* estimation was performed according to the authors’ own methodology using an LMI 4003 plastometer (DYNISCO Polymer Test, Rochester, NY, USA). The polymer sample melted at 180 °C was extruded through a spinneret with a 0.5 mm capillary, at a piston load of 2.16 kg. The weight of the polymer extruded in a defined time (10 min) was determined, and the MFR was calculated. The filament obtained during the analysis was used for metrological tests in order to assess the influence of the plasticizers on the mechanical properties of PLA.

*Molar mass distribution and polydispersity* of samples were analyzed by the SEC-MALS method. The tests were performed using an Agilent 1200 Series (Agilent Technologies, Santa Clara, CA, USA) equipped with an Optilab refractometric detector (Wyatt Technology, Goleta, CA, USA) and an 18-angle MALS HELEOS light scattering detector (Wyatt Technology, Goleta, CA, USA). The tests were performed using chloroform as the eluent and two PLgel Mixed-C 300 × 7.5 mm chromatographic columns (Agilent Technologies, Santa Clara, CA, USA) at a flow rate of 1 mL/min.

*Mechanical properties* were determined at standard environmental conditions (20 ± 2 °C and RH 65 ± 4 %) according to the PN-EN ISO 139:2006 standard. The following parameters were estimated using an Instron 5544 tensile testing machine (Norwood, MA, USA):-linear density (segment method) according to PN-P-04653:1997-breaking tenacity according to PN-EN ISO 2062:2010 method A-breaking force and elongation at break according to PN-EN ISO 2062:2010 method A-initial tensile modulus according to PN-P-04669:1984

The filaments obtained using LMI 4003 plastometer during the MFR analysis were used for metrological tests in order to assess the influence of the plasticizers on the mechanical properties of PLA.

*Structural analysis* was carried out by the Fourier transform infrared spectroscopy-attenuated total reflectance (FTIR-ATR) method using a Nicolet iS50 Spectrometer (Thermo Scientific, Waltham, MA, USA). The operating parameters were as follows: measurement range 4000–400 cm^−1^, resolution 4.0 cm^−1^, and the number of scans for baseline and spectrum collection was 32 using detector DTGS ATR. Samples for tests were prepared in the form of films, which were prepared by a casting method using chloroform as a solvent. The accuracy of wavenumbers reading for characteristic bands was ± 1 cm^−1^.

*Scanning electron microscopy (SEM)* investigation of the PLA regranulates was carried out using a Quanta 200 scanning electron microscope (FEI, Hillsboro, OR, USA). Tests were carried out on gold sputter-coated samples in a high vacuum at an electron beam accelerating voltage of 5 KV and magnification of 100× and 1000×.

## 3. Results and Discussion

### 3.1. Characteristics of the Modified Regranulates

For base PLA and PLA regranulates containing 10 wt.% of plasticizer, thermal analysis was performed using the DSC technique. The results are presented in Table 2 and Figure 1.

Analyzing the results obtained from the thermal tests (Table 2), it can be seen that the glass transition temperature (T_g_) for all modified samples was shifted towards lower values compared to the base polymer (Figure 2). For samples 2, 3, 4, and 5, T_g_ values were below 40 °C. The lowest T_g_ value of 34 °C was observed for the regranulate with ethoxylated stearic acid (sample 3). A single glass transition (one T_g_ value) was observed for all modified samples, which proves the good miscibility of the components of the prepared compositions. The plasticizers used in samples 1, 2, and 3 did not cause significant changes in the melting temperature (T_m_), while for tests 4, 5, and 6, a double transition to the plastic state was observed, as evidenced by two T_m_ values (Tabele 2). In the tested samples, however, there were differences in the cold crystallization temperature (T_cc_) and the cold crystallization enthalpy (∆_cc_), which indicates the low-ordered structure. A slight increase in the degree of crystallinity of the modified samples was observed in relation to the completely amorphous starting sample (Figure 1). The degree of the structure order is also reflected by the value of the crystallization enthalpy (∆H_c_). The applied plasticizers cause a slight increase in the order of the regranulate structure, which is particularly noticeable for samples 2 and 3.

The regranulates obtained after modification with plasticizers were subjected to rheological analysis by determining the MFR at 180 °C (Figure 3). For all the modified samples, there was an increase in the MFR in relation to the initial sample. The highest MFR values were observed for samples 2 and 3, also the lowest T_g_ values were determined for these samples (respectively 34.9 °C and 34.1 °C).

### 3.2. SEC-MALS Analysis

For the base PLA and plasticized regranulates, SEC-MALS analysis was performed to determine the weight-average molar mass (*M_w_*) and the molar mass distribution (MMD). The results are presented in Figure 4 and Figure 5. The results show the absolute values, which were obtained using the MALS detector.

For regranulates 1, 2, and 3, after the plasticization in an extruder, a decrease in the value of *M_w_* was observed, respectively, by 24.9, 40.2, and 38.6% in relation to the base polymer. Samples 4, 5, and 6 after plasticization did not change their molecular weight distribution in relation to the base PLA sample, which indicates their molecular stability during processing. The molar mass decrease in samples 1, 2, and 3 was accompanied by an increase in MFR (Figure 2). The effect of the lower *M_w_* value is lower viscosity of the polymer in the plastic state, which results in faster melt flow. In samples 1, 2, and 3 the plasticizers did not reduce the interactions between the polymer macromolecules, polymer degradation occurred which was reflected by a shift of the MMD curves towards lower molar mass values. For samples 4, 5, and 6 no polymer degradation was observed and the MFR value increased by over 170%, in relation to the base PLA sample which is evidence of the influence of the plasticizers used on the PLA macromolecules.

### 3.3. Mechanical Testing

For the filament produced from sample 2 plasticized using block copolymer of ethylene oxide and propylene oxide, it was not possible to perform mechanical tests due to the excessive brittleness of the sample which broke during clamping in the tensile testing machine. For the other samples in the form of filaments mechanical testing was performed and the results are presented in Figure 6, Figure 7, Figure 8 and Figure 9. The filaments obtained using LMI 4003 plastometer during the MFR analysis were used.

For all tested samples, a decrease in the linear density was observed in comparison to the base PLA. For sample 4 the linear density was reduced by 21% and for samples 5 and 6 by 15%. For samples 1 and 3 there was a significant decrease in linear density, respectively by 36% and 52%, also for these samples a decrease in *M_w_* and MFR was observed.

Modified samples 3 and 6 showed only slightly lower values of breaking tenacity as compared with base PLA (approx. by 13%), while for samples 1, 4, and 5, a significant decrease in this parameter was observed. Respectively, for samples 1 and 4, the decrease in tenacity was approx. 33% and for the sample 5 was 56%. Based on the results, the elongation at break increased significantly only for samples 4 and 5 and was 21% and 35%, respectively wherein both used plasticizers had a very similar chemical structure. The increase in the breaking elongation shows the effect of the plasticizers used on the polymer.

Young’s modulus, which is one of the basic properties characterizing a given material, was also determined. It is the coefficient of proportionality between the normal stress and the elongation. The test results are presented in Figure 8. For all the modified samples, lower values of the Young modulus were observed compared to the base PLA sample. The lowest values of approx. 180 cN/tex were recorded for samples 4 and 5, for which the plasticizers belonging to the same group were used. A decrease in the Young’s modulus in samples 4 and 5 means that the samples are more flexible and are easily subjected to stretching (deformation), which is also visible in the measured values of elongation at break.

### 3.4. FTIR-ATR Analysis

For base PLA and PLA regranulates containing 10 wt.% of plasticizer, structural analysis was performed using the FTIR-ATR technique. The results are presented in Figure 10. The spectral analysis was also performed for the plasticizers (TEC, ADO, SDO), for which changes in the plasticized polymer were observed in the FTIR-ATR spectra (Figure 11).

As a result of the modification of PLA with plasticizers, structural changes in the FTIR-ATR spectrum (Figure 10) were observed. The first difference in the high wavenumber region of 3000–2800 cm^−1^, characteristic for the C-H stretching vibration was observed. Plasticization with ADO and SDO (samples 4 and 5) caused disturbances for bands around 2870 cm^−1^ (asymmetric stretching of C-H in the methyl group -υ_as_CH_3_) and 2960 cm^−1^ (symmetric stretching of C-H in the methyl group-υ_s_CH_3_). For the remaining modifiers, this phenomenon was not observed [50]. In the spectrum (Figure 10—No. 1) peaks form plasticizers were observed (Figure 11—No.1).

The second difference in region characteristic for –C=O ester carbonyl group around 1750 cm^−1^ was observed. This peak (Figure 10—No 2) was deformed only after modification by ADO and SDO. In the spectrum of plasticizers (Figure 11—No.2) the peak around 1730 cm^−1^ for the ester group was observed [51]. For plasticized PLA by ADO (sample 4) and SDO (sample 5), the deformation in the range of 1750–1760 cm^−1^ was observed. It may indicate that this group also interacts with plasticizers. The third difference in the FTIR-ATR spectrum concerned all samples after plasticization (Figure 10—No. 3.). The FTIR-ATR spectrum of the base PLA film showed the complete lack of the band at 923 cm^−1^, which is assigned to the coupling of the C-C backbone stretching with the CH_3_ rocking mode and sensitivity to chain conformation of PLA crystals [52,53,54]. This study also confirmed the DSC analysis.

### 3.5. SEM Analysis

After modification, the regraulates were assessed using a scanning electron microscope (SEM) to observe the changes on the surface. The results are presented in Figure 12. No significant changes on the surface of the granules in the SEM pictures for samples 2, 4, 5, and 6 were observed. For modified samples 1 and 3, only a slight surface roughness was observed.

### 3.6. Thermal Gravimetric Analysis

For the base PLA and plasticized regranulates, TG analysis to determine the decomposition temperature of samples was performed. The results are presented in Figure 13 and Table 3.

Modified samples 4, 5, and 6 showed only slightly lower decomposition temperatures (T_1_, T_max_, T_2_); also the weight loss is at the same level as the base PLA. The lowest values TG parameters were recorded for samples 1, 2, and 3, for which the plasticizers belonging to the same group were used (macromolecular compounds -nonionic surfactant, containing ether bonds). For these samples, polymer degradation was observed because the lower molar mass was determined. The degradation of the polymer accelerated the thermal decomposition. The optimal processing temperature for PLA and plasticized regranulates is in the range of 200–220 °C. According to TG analysis, the effective start temperature of the mass loss process is more than 300 °C and it provides a sufficient temperature buffer.

## 4. Conclusions

For all modified samples the glass transition temperature, depending on the plasticizer used, was shifted towards lower values compared to the base polymer. The samples 2, 3, 4, and 6 showed T_g_ values below 40 °C. The lowest T_g_ value at the level of 34 °C was obtained for the regranulate plasticized with ethoxylated stearic acid (sample 3). Macromolecular plasticizers containing ether bonds cause polymer degradation during the regranulation process, as evidenced by the decrease in *M_w_* value, MMD shift towards lower fractions, and higher MFR index. According to MSDS issued by the PLA manufacturer, processing at 220–240 °C is recommended. Unfortunately, the plasticizers used are subject to degradation above 200 °C. It was not possible to lower the processing temperature below 200 °C as this could damage the processing equipment due to the presence of not fully plasticized polymer. The temperature in the range of 200–220 °C was used in the plasticization tests. As a result of the degradation of the plasticizer (ether bonds), water molecules appeared, which also resulted in the degradation of the polymer [37]. During planning the research it was assumed that some part of the plasticizer may degrade. However, it was not expected that degradation of plasticizer would affect the structure of PLA so much, causing the degradation of the polymer. It was expected that the degradation of the plasticizer and the appearance of smaller molecules would reduce the intermolecular interaction according to the free-volume theory.

The plasticizer used in sample 2—block copolymer of ethylene oxide and propylene—caused a decrease in T_g_ value and an increase in MFR as a result of degradation of PLA. It did not improve the plastic properties of the polymer, and on the contrary, it caused high brittleness of the filament which broke during clamping in the tensile testing machine.

Plasticizers used in sample 4 (di-2-ethylhexyl adipate) and sample 5 (di-2-ethylhexyl sebacate) affect PLA and reduce its intermolecular interactions. This is confirmed by the tests performed, in which for both samples the lower T_g_ value was determined, respectively 39.7 °C for sample 4 (ADO), and 48.1 °C for sample 5 (SDO). In both cases, the MFR coefficient was determined at the level of approx. 1 g/10 min (for base PLA 0.36 g/10 min), but no changes in the molar masses were observed. Additionally, lower values of Young’s modulus of approx. 180 cN/tex in relation to the base PLA sample (244 cN/tex) and significant values of the elongation at break were recorded, respectively 20% for sample 4 and 34.6% for sample 5. ADO and SDO plasticizers showed a better effect on PLA compared to citrate (sample 6) recommended in the literature. For ADO and SDO at the FTIR-ATR spectrum were observed deformation of peaks, which may indicate the interaction of ester carbonyl group of PLA with plasticizers. The structural change was not observed for other plasticizers at the FTIR-ATR spectra except crystallinity change. According to the free-volume theory, the structure of the plasticizer is important. Based on the research it can be seen that the increase of the molar mass plasticizers determined the increase of the elongation according to free-volume theory, for SDO (the bigger molar mass from low-molecular plasticizer) the highest elongation was obtained. On the other hand, it was not observed that the elongation increased with the decrease of the glass transition temperature (TEC—35 °C, ADO—39 °C, SDO—48 °C), the glass transition temperature is one of the most important aspects of plasticization according to the free-volume theory. In fact, the process is more complicated than this theory does not consider e.g., the phenomenon of anti-plasticization, other properties such as viscosity or elastic modulus also associated with plasticization. Because the plasticization of polymers is a process dependent on many factors (e.g., chemical structure of the plasticizer, polymer structure, amorphous phase content, crystal phase structure, molar mass, and polymer polydispersity) for each polymer and application an appropriate plasticizer should be selected to achieve an effective plasticization. The selected plasticizers ADO and SDO are used in the production of food contact materials, in particular in the production of PVC food films, also exhibited great potential to be applied to PLA food films, and exhibit better properties than the citrate, which are indicated in many publications as PLA plasticizers.

Because the EU is focused on green technology solutions, biopolymers as PLA have become noticeable by producers from various industries. Despite the higher price of PLA than traditional polymers, the social and environmental aspect becomes more important than the economic cost. The UE Directive 2008/98/EC on waste introduces the “polluter-pays principle” whereby the original waste producer must pay for the costs of waste management. The important concept of extended producer responsibility (EPR) is introduced in the directive and it induces us to look for new solutions using biodegradable polymers. Therefore, PLA may become an alternative that fits in with the circular economy and the system EPR. Because of the limitations of this material, research on modification and changes in its properties will be important in the coming years. The research presented in this paper shows how the elastic properties of PLA can be improved. It opens up new opportunities for industrial that were previously unattainable. Additionally, consumer awareness is now at a higher level, and the desire to buy food in bio-based packaging is growing.

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
