# Peer review of "Modification of Poly(lactic acid) by the Plasticization for Application in the Packaging Industry"

_polymers, 2021, doi:10.3390/polym13213651_

Round 1
Reviewer 1 Report
Comments listed below to authors should address properly:
- The introduction should improve by avoiding the general information in the beginning of introduction and should including more information to clarify the importance of this topic, which should cite from related works literature.
- Authors should include a schematic representation of the formation of PLA composite.
- TGA test is recommended to be tested to show the improvements of thermal stability for composites.
- The quality of the manuscript can be improved by giving the SEM images of the PLA composites. Authors should include SEM images for some PLA samples.
- Discussion of the results is not enough and should improve by interpreting and clarify the significant presents in detail and should compare with previous work to show the difference in improvements.
- Tensile stress-strain curves should include in the mechanical test section to show the tensile strength and modulus for PLA composites.
- Figure 2: DSC curves must be changed by individual figures. Also, there are no values visible in the Y axis showing the temperature values range.
- Authors should present FTIR spectra in Figure 10 with high resolution figures and they must be shown clearly.
Author Response
Dear Reviewer,
I'd like to thank you for a helpful review.
I will try to correct the article in accordance with the reviewer's guidelines.
I'll correct the too general introduction. I will try to make it more introductory.
I’ll prepare a schematic representation of the formation of PLA composite ,and SEM and TG analysis will be added. I will change DSC curves for individual and the FTIR pictures will be improve.
Best Regard
Karolina Gzyra-Jagieła
Reviewer 2 Report
Dear Authors,
The manuscript submitted for review requires a few corrections. They mainly concern the introduction. The research methodology also requires some explanations.
Detailed comments below:
I must admit that this introduction is very general. The presented information about PLA can be found in many articles in this field. In my opinion, you should show the advantages of PLA against the imperfections of other biodegradable products. This will raise the validity of your research. Unfortunately, PLA is a highly processed raw material, usually produced from corn (a food product). Therefore, increasing the scale of PLA applications must be really justified.
Each acronym must be described before first use. Review all work.
Line 92-104. This part of the work is the research methodology. Such information should therefore be included in the "Materials and Methods" section.
Line 129: Basically the rest (up to line 139) is the purpose of your research and its rationale. Such information should therefore be included at the end of the introduction. The methodology should contain specific information on the material, apparatus and methods of scientific research.
Line 141: Add a symbol for this extruder. In addition, enter the length-to-diameter ratio of the L / D screws.
Line 144: Then you use the unit "tex". This unit is for thread (textile). In this case, you write about granules?
Justify the use of this unit in the methodology for testing strength parameters.
Generally, the basic unit for tensile stress testing is MPa.
Line 245: Minor y-axis mistake. (Liner?)
Line 268: cN - is it correct? I understand you meant kilonewton kN.
Line 304: Add some more forward-looking conclusion. A conclusion that will show the perspective of your research.
Author Response
Dear Reviewer,
I'd like to thank you for a helpful review.
I will try to correct the article in accordance with the reviewer's guidelines.
I described all acronym before first use.
Line 92-10 in the "Materials and Methods" section was transferred. Lines 129-139 at the end of the introduction were transferred. I Added a symbol for extruder and the length-to-diameter ratio of the L / D screws.
The filaments obtained using LMI 4003 plastometer during the MFR analysis was used for metrological tests in order to assess the influence of the plasticizers on mechanical properties of PLA. it was included in the "Materials and Methods" (line 184) but I added it also in line 200 and 273. Because filaments are used in mechanical testing, therefore ‘tex’ and ‘cN/tex’ (centy Nuton, 1 [cN] = 0,01 Niuton [N]) is used. Due to the equipment (INSTRON) I have, this was the only way I could examine the mechanical properties. However, on the basis of the mechanical analysis of the filaments, the differences and modification possibilities are clearly visible.
Best Renards
Karolina Gzyra-Jagieła
Round 2
Reviewer 1 Report
-
Reviewer 2 Report
I accept the introduced changes. In my opinion, the article can be published as is.